# How Nitric Oxide Hindered the Search for Hemoglobin-Based Oxygen Carriers as Human Blood Substitutes

**DOI:** 10.3390/ijms241914902

**Published:** 2023-10-04

**Authors:** Michele Samaja, Ashok Malavalli, Kim D. Vandegriff

**Affiliations:** 1Department of Health Science, University of Milan, 20143 Milan, Italy; 2Vivosang Inc., Dover, DE 19901, USA; asmalavalli@vivosang.com (A.M.); kdvandegriff@vivosang.com (K.D.V.)

**Keywords:** hemoglobin, blood substitutes, heme, nitrosylation, nitrosothiolation, vasoconstriction, red blood cells, oxygen carrying, NO scavenging

## Abstract

The search for a clinically affordable substitute of human blood for transfusion is still an unmet need of modern society. More than 50 years of research on acellular hemoglobin (Hb)-based oxygen carriers (HBOC) have not yet produced a single formulation able to carry oxygen to hemorrhage-challenged tissues without compromising the body’s functions. Of the several bottlenecks encountered, the high reactivity of acellular Hb with circulating nitric oxide (NO) is particularly arduous to overcome because of the NO-scavenging effect, which causes life-threatening side effects as vasoconstriction, inflammation, coagulopathies, and redox imbalance. The purpose of this manuscript is not to add a review of candidate HBOC formulations but to focus on the biochemical and physiological events that underly NO scavenging by acellular Hb. To this purpose, we examine the differential chemistry of the reaction of NO with erythrocyte and acellular Hb, the NO signaling paths in physiological and HBOC-challenged situations, and the protein engineering tools that are predicted to modulate the NO-scavenging effect. A better understanding of two mechanisms linked to the NO reactivity of acellular Hb, the nitrosylated Hb and the nitrite reductase hypotheses, may become essential to focus HBOC research toward clinical targets.

## 1. Introduction

The search for a reliable substitute of blood in the transfusion setting is still ongoing in an exhaustive manner to cover an urgent need in modern society [1]. After dismissing perfluorocarbon-based technology after decades of work [2], and before targeting approaches based on induced pluripotent stem cells that do not yet appear mature for clinical use [3], efforts are presently concentrating in hemoglobin (Hb)-based oxygen (O_2_) carriers (HBOC). Although HBOCs have been investigated for more than 50 years and several reviews have discussed them as potential candidates [1,4,5,6,7,8], up to now, no affordable alternatives to allogeneic blood transfusion have yet emerged, at least in acute contexts linked to hemorrhagic shock, where replacing the blood loss without compromising the O_2_ carrying function is the primary goal [9]. Among the plethora of side effects that have emerged from time to time, the fast reaction of acellular Hb with nitric oxide (NO) represents a serious and yet unsolved mechanism that has prevented and is still preventing the launch of affordable HBOCs.

NO was long believed to be simply a small gaseous molecule that binds the heme core Fe^2+^ atom with a higher affinity than that of O_2_. Having a higher affinity for Hb than O_2_, even higher than carbon monoxide (CO), NO appears to be a toxic gas because it negatively interferes with the Hb’s O_2_ carrying function. Later, NO was identified as the potent endothelium-derived relaxing factor that, besides other functions (Figure 1), controls, via the NO-cyclic guanosine monophosphate (cGMP) pathway, the vasoactivity in health and disease [10]. The bioactivity of NO in circulation is regulated by factors that include various NO synthases. In circulation, the endothelial isoform or eNOS is not only preponderant, but also the most critical in vivo [11]. NO bioactivity is regulated through the modulation of the enzymatic pathways that carry on the signal originated by NO, as the various inhibitors of phosphodiesterase (PDE), the conversion of NO in non-vasoactive inorganic forms as nitrites (NO2−) and nitrates (NO3−), and the presence of NO scavengers, that include heme proteins and Hb. The modulation of the enzymes that carry on the signal originated by NO as the various inhibitors of phosphodiesterase (PDE), the conversion of NO in non-vasoactive inorganic forms as nitrites (NO_2_^−^) and nitrates (NO_3_^−^), and the presence of NO scavengers, involves heme proteins and Hb. When Hb resides in the intact red blood cell (RBC), the NO-scavenging effect is finely regulated to augment O_2_ delivery to hypoxic tissues through the interaction of at least three mechanisms: (1) the barrier constituted by the RBC membrane [12]; (2) the binding of NO to the β93Cys residue in human Hb [13] followed by the intramolecular transfer of NO from the β93Cys to the heme, favored by low levels of competitive heme binding by O_2_ (i.e., local hypoxia); and (3) the human Hb nitrite reductase activity [14] that reversibly transforms vasoactive NO into non-vasoactive inorganic forms. Once released in the circulation, free NO targets soluble guanylate cyclase (sGC) in the smooth muscle cell (SMC), where it triggers a pathway leading to relaxation that increases the capillary diameter, thereby favoring O_2_ delivery to tissues. NO can also target platelets, favoring their inactivation. We will discuss how this chain of events is severely challenged when part of Hb resides in the plasma, a situation where the diffusion distance between NO and acellular HBOCs is decreased, as always happens following HBOC infusion.

Deciphering the above mechanisms would not only accelerate the search for clinically usable HBOCs but may also provide a unique opportunity to understand the details of the role of NO in human biology. Thus, the purpose of this article, in the Special Issue dedicated to the 25th anniversary of NO, is not to add a further review of past and present HBOC candidates, but rather to focus on the basic biochemical and physiological events that hindered the launch of a single HBOC candidate in the clinical arena, with special concern to one of the several interactions of Hb with other molecules: its reaction with NO. We limited our search of the literature to peer-reviewed articles, excluding abstracts, patents, book chapters, and other non-reviewed reports. We apologize for not citing several articles for space reasons.

## 2. NO Scavenging: The Difficult Co-Existence of NO with Hb

### 2.1. The Reaction of Hb with NO

NO binds the Hb heme with an affinity greater than that of O_2_ and even of CO [15]. The bimolecular rate constant for the association of deoxygenated Hb with NO to form nitrosylated Hb (HbNO), as measured in rapid mixing devices, is of the order of 10 μM^−1^ s^−1^ [16], which enables the completion of the reaction in approximately 20 ms under quasi-physiological conditions. These values remain roughly similar for most modified human and bovine deoxygenated Hb derivatives, including those intramolecularly cross-linked, intermolecularly cross-linked polymerized, and conjugated with polyethylene glycol (PEG) [17]. In arterial blood, both RBC and acellular Hb are in the oxygenated form, which translates into a more complex scenario because of two additional phenomena: the binding of heme with NO is limited by the rate of O_2_ dissociation from oxygenated Hb (HbO_2_), and the simultaneous presence of O_2_ and NO oxidizes ferrous Hb to ferric Hb or methemoglobin (Hb^+^) unable to bind any gas except NO. The bimolecular reaction rate of 20 μM^−1^ s^−1^ for the NO-driven Hb oxidation is, however, of the same order of magnitude as the bimolecular association of NO to deoxygenated Hb [18]. This reaction is more complex than it appears because of the formation of a peroxynitrito-complex Hb^+^OONO that decays rapidly to Hb^+^ and NO_3_^−^ [19]. The mechanism is further complicated by the fact that, unlike O_2_ and CO, NO can bind Hb^+^, forming an unstable HbNO^+^ complex. Severe pro-oxidant conditions may generate a reactive ferrylHb derivative that is reduced by NO to Hb^+^ and NO_2_^−^ [20].

When intact RBCs are in an O_2_-rich environment such as arterial blood, the competition with O_2_ for the iron heme redirects NO to bind human Hb at a strictly conserved Cys residue in position β93 (β93Cys) to form the nitrosothiolated derivative HbSNO. The underlying molecular mechanism is supported by crystallographic studies showing that β93Cys is highly reactive in HbO_2_, but unreactive in deoxygenated Hb [21]. This feature enables the consideration of Hb not simply as a NO scavenger, but as a transporter of NO to O_2_-poor tissues. In fact, the increased fraction of deoxygenated Hb favors the intramolecular transfer of NO from β93Cys to the heme, with competitive binding of NO to the RBC membrane band 3 anion exchanger that exports NO outside the RBC into the bloodstream to increase the flow of blood and O_2_ [13]. This mechanism may be particularly relevant in local hypoxia in the presence of greater fractions of deoxygenated Hb. The availability of knock-in humanized mice carrying the β93Cys→Ala mutation might validate this hypothesis, but has given controversial results. On the one hand, hypoxia-regulated HbSNO-mediated vasodilation has been confirmed in vivo, despite numerous compensations in animals carrying the β93Ala mutation to alleviate tissue hypoxia [22]. On the other hand, no appreciable differences between β93Cys and β93Ala mice were detected in several models, which included the resistance to myocardial ischemia/reperfusion, the export of NO bioactivity in human platelets co-incubated with RBCs, and the vascular reactivity in aortas pre-treated with S-nitrosoCys, hemolysates or HbNO, as well as the hypoxic vasodilation in dorsal skin-fold windows [23]. Likewise, no differences in systemic and pulmonary hemodynamic were detected under hypoxia between two groups of mice [24]. These observations apparently do not support the relevance of HbSNO-based mechanisms from the O_2_ offloading perspective, despite greater ROS formation in β93Ala vs. β93Cys RBCs and lungs that, nevertheless, highlights a critical role for β93Cys in the antioxidant network [25].

Although this mechanism would be central to enable Hb to release NO depending on the hypoxia state or Hb-O_2_ saturation, there are arguments against this hypothesis based on both the stoichiometry of Hb binding with NO and on the RBC environment. First, the formation of HbSNO in vitro has been questioned due to technical artifacts attributed to the very fast binding of NO with HbO_2_ [26]. Second, similar rates of NO binding to HbO_2_ and Hb indicate its independence to the Hb-O_2_ saturation [27,28]. Third, the transnitrosation from HbSNO to the RBC membrane band 3 anion exchanger, which exports NO to the circulation, may be too slow to take place within the time spent by RBCs along the capillary [29]. Fourth, increased Hb-O_2_ affinity in HbSNO implies that Hb must flow through environments at a very low PO_2_ in order to be deoxygenated enough to act as a NO transporter [29]. Thus, the role of Hb as a NO scavenger or NO transporter in vivo is at least controversial, and accurate measurements of HbSNO stability in arterial and venous blood are required [30]. Also, the possible reaction of Hb^+^ with NO may perhaps be pivotal to understand the complex underlying mechanism [31].

### 2.2. RBC Hb and NO Scavenging

As already pointed out [32], when Hb resides in the intact RBCs, the reaction of NO with Hb is hampered by the presence of at least four barriers that create diffusion constraints to any gas: (1) an RBC-free zone adjacent to the endothelium due to the presence of a laminar flow zone; (2) the diffusion of NO to the RBCs, hampered by the presence of an unstirred layer around the RBCs; (3) the RBC membrane; and (4) intracellular NO diffusion. This situation is profoundly modified in the case of hemolysis that removes part of the diffusion constraints and considerably accelerates the reaction of Hb with NO, perhaps by a factor of 1000 [33,34], considerably reducing NO bioavailability in the circulation [14]. In principle, the infusion of HBOCs represents a situation like hemolysis, with acellular Hb in the plasma that scavenges circulating NO faster than RBC Hb, thereby leading to vasoconstriction and hence hypertension [35,36]. More specifically, the rapid dioxygenation of NO catalyzed by acellular Hb forms several species that include NO_2_^−^ and NO_3_^−^ [37], HbSNO [38], and HbNO [39]. Although these molecules are per se non-vasoconstrictive, they prevent NO diffusion to SMCs with consequent capillary contracture. A detailed mathematical model, developed to quantify the resistance to NO transport around a single RBC and to predict the NO consumption rate in the presence and in the absence of acellular Hb, predicted that the difference in NO uptake by RBCs and free Hb is smaller than that reported and is hematocrit-dependent [40]. Nevertheless, the presence of acellular Hb in circulation has been predicted to reduce the NO level in the SMC below the activation range (12–28 nM) for sGC, a major determinant of SMC contraction [41]. Furthermore, this concentration may be significantly reduced in the case of HBOC extravasation, due to the presence of myoglobin that scavenges NO and further reduces the SMC NO level [41].

### 2.3. NO Signaling

Traditionally, NO-cGMP signaling targets SMC sGC to reduce intracellular Ca^++^ levels negatively modulating the activity of myosin light-chain kinase (MLCK), which results in SMC relaxation, greater recruitment of the capillary network, and increased local blood flow. The involvement of the NO-cGMP pathway in NO scavenging by acellular Hb has been demonstrated in isolated Langendorff-perfused rat hearts pharmacologically preconditioned with inhibitors of the PDE isoform 5, most abundant in SMC, which suppresses vasoconstriction [42]. However, the frequency of SMCs in terminal capillaries and brain tissue decreases progressively, which enables switching the control of blood flow to endothelial cells and pericytes, which contain most but not all of the machinery found in SMCs [43]. This hypothesis was refuted by the finding that pericytes lack the contractile protein actin and play no role in squeezing blood vessels [44], but another study showed that blood flow in the brain is exclusively controlled by SMCs surrounding large arterioles, with others reporting α-actin in pericytes, which enables them to regulate blood flow [45]. There is evidence that the NO-cGMP signaling cascade mediates pericyte contraction in the same way as it does in SMC [46].

In addition to the above signaling cascade, NO has direct effects on the cellular bioenergetic metabolic flux at the mitochondrial level. In competition with O_2_, NO, or more likely the peroxynitrite derivative (ONOO^−^) [47], markedly decreases the activity of cytochrome c oxidase, which impairs the mitochondrial electron flux [48,49] and regulates superoxide production, especially at low PO_2_ [47]. The mitochondrial dysfunction caused by NO may have lots of downstream effects, including the activation of protein kinase signaling with the modulation of sterol regulatory element-binding protein 1, acetyl-CoA carboxylase, medium-chain specific acyl-CoA dehydrogenase, and mitochondrial and peroxisome proliferator-activated receptor-γ coactivator 1α [50]. In several models, including certain types of tumors [51,52,53], healthy brain [54], and lungs with pulmonary hypertension [55], NO-driven metabolic reprogramming towards aerobic glycolysis, or the Warburg effect, showed protective effects. Remarkably, this metabolic reprogramming is quite similar to that observed in the immune system and in endothelium [56]. In macrophages, the activation of the Warburg effect exerts anti-inflammatory effects [57].

### 2.4. NO Signaling in the Presence of HBOCs

In the presence of acellular Hb in circulation, Hb-driven NO scavenging emerges as a critical phenomenon that suddenly removes the vasodilatory effects of NO. The critical role of NO scavenging in leading vasoconstriction was confirmed by testing the hypothesis that inhaled NO may mitigate the vasoconstrictive effects of HBOCs [58]. Indeed, inhaling 80 ppm one hour before HBOC infusion in mice, lambs, and sheep could prevent most of the deleterious effects of NO scavenging [58,59,60]. This treatment appeared to be effective also in a single compassionate use in humans [61]. There are at least two instances where the chain of events described above is attenuated, indicating that the classical NO-cGMP signaling pathway may not be sufficient to cover all the mechanisms underlying the effects of NO in circulation.

First, a study measuring the arteriolar diameter in hamsters infused with various types of HBOCs using intravital microscopy demonstrated that, despite equivalent reductions in the NO concentration with each HBOC, PEGylated Hb was associated with the maintenance of baseline arteriolar diameter, while for other HBOCs as α-α cross-linked and polymerized Hb, the vasoconstriction was observed as expected [62]. A plot of the arteriolar diameter against the perivascular NO concentration illustrates that PEGylated Hb decoupled the decreased NO concentration from vasoconstriction, as indicated by the vessel diameter. Although the vessel diameter is not necessarily an expression of tissue oxygenation, the observed differences might indicate that PEGylated Hb exhibits a lack of vasoconstriction that could not be explained simply by the differences in NO scavenging common in most Hb solutions [17].

This observation points to the alternative hypothesis that the control of vasoactivity may be ascribed, at least in part, to O_2_ within an autonomous drive aimed at matching the O_2_ delivery to the tissue O_2_ demand, as observed in rat skeletal muscles [63], and indirectly confirmed by the vasoconstriction often observed in hyperoxic brain [64], heart [65], and skeletal muscle [66], along with various vascular beds [67], but not in kidneys [68] nor lungs [69]. The underlying mechanisms may involve the control exerted by O_2_ on arteriole constriction [70]. The observation that human adult Hb and a Hb derivative with a high affinity for O_2_ (low p50) display similar rates of NO association to deoxygenated, but not oxygenated Hb, further confirms that the NO mechanisms are not the only driver for vasoactivity in physiological conditions [18].

### 2.5. Nitrite Reductase Activity as a Mechanism of Vascular Control

One potential mechanism for explaining the diminished vasoconstriction observed with various kinds of HBOCs is associated with enhanced NO regeneration in circulation through the reduction of endogenous NO_2_^−^ by heme proteins [71]. The postulated reaction scheme shown in Figure 2 may explain how the NO regeneration potential overrides with certain types of HBOCs, and the high rate of NO association with Hb in both the oxygenated and deoxygenated forms, i.e., the physiological case. This favors vasodilatation and regenerates NO_2_^−^.

## 3. Main Problems Associated with HBOC Infusion: A View from NO

### 3.1. Systemic Effects

A large number of clinical trials have addressed numerous safety concerns related to HBOC infusion, mostly regarding hypertension and injury to the cardiac, renal, gastrointestinal, pancreas/liver, and central nervous systems [74]. Most of these adverse effects were attributed to the NO-scavenging effect, that, along with hyperoxygenation and heme toxicity, leads to the rapid onset of hypertension [75]. In hearts, this increases the incidence of myocardial infarction in compromised patients [76], but the function of almost every organ is compromised by HBOCs. However, brain and cerebral tissue dysfunction are perhaps the most critical. In fact, many of the defenses orchestrated by the body to fight external stress converge into the preservation of brain function. For example, in swine exchange-transfused with polymerized Hb, the marked changes in systemic and pulmonary pressures were not accompanied by measurable consequences in brain circulation, as an outcome of the effort of the organism to protect the brain at the expense of dysfunction in other systems [77]. Indeed, the capillary network in brain tissue possesses efficient mechanisms that enable the autoregulation of blood flow to protect oxygenation when the brain is challenged by hemorrhagic shock [78]. Likewise, cerebral blood flow is uncoupled to the metabolic O_2_ uptake [79], unlike almost all other tissues. The exchange transfusion of α-cross-linked Hb in rats led to a marked increase in the hypoxia-related response, which translated into irreversible neuron apoptosis and death [80]. The observation timings were unable to detect potentially protective hypoxic conditioning effects, but NO regeneration via enhanced nitrite reductase activity [81,82], with the stimulation of defense mechanisms such as Nrf2, ERK1/2 and iNOS, may reduce neuron death over longer times [83].

### 3.2. Inflammation

An additional systemic effect linked to acellular Hb is the development of inflammation and the modulation of the immune response [84]. NO is well known to play a key role in these phenomena in a biphasic manner. Although their overproduction has pro-inflammatory effects, NO and ROS exert multiple effects in the regulation of immune responses in virtually every step of inflammation development [85]. The low NO levels caused by decreased NO production by the eNOS pathway are known to cause inflammation of the vascular endothelium [86,87], with an inhibited expression of adhesion molecules, cytokines and chemokines, and impairment of the body’s defense against invading microorganisms. The same outcome is expected when NO levels are low because of NO scavenging by HBOCs. However, some recent HBOC products have shown positive features concerning inflammation. For example, glutaraldehyde-polymerized bovine Hb exhibited a significant reduction in tissue injury and neutrophil infiltration in a canine model of myocardial ischemia-reperfusion injury [88]. Perhaps this effect is secondary to the HBOC-induced overexpression of heme oxygenase-1 in hearts and lungs, but not brains [89]. The same results were found in liver Kupffer cells [90]. The use of polynitroxylated Hb derivatives markedly inhibits free radical-induced microcirculatory dysfunction [91].

### 3.3. Coagulopathy

Both endothelium and platelet-derived NO prevent platelet adhesion to the vessel wall and inhibit thrombi growth. The NO-cGMP pathway in platelets targets the phosphorylation of the thromboxane A2 receptor (TXA2), which prevents platelet activation, shifting the pro- vs. anti-coagulation balance toward clot dissolution [92]. In several cases, such as during extracorporeal support, NO administration is a valid alternative to heparin-based therapy [93]. NO also suppresses the production of the inhibitor of the plasminogen activator in vascular endothelial cells, in antagonism with vasoconstrictive substances such as endothelin-1 [93]. The pharmacological administration of NO donors enhances the anticoagulant factors in a dose-dependent manner by building the nitrite pool [94]. Thus, Hb-driven NO scavenging would appear to trigger a potentially lethal hypercoagulability state. However, as a matter of fact, the literature data on the coagulability state after HBOC infusion are not definitive. Despite evidence of platelet aggregation driven by free Hb [95], most studies did not highlight a clear coagulopathy associated with NO scavenging in rats transfused with polymerized Hb at 50% dilution [96], and in swine exposed to hemorrhagic shock and transfused with bovine polymerized Hb [97,98]. In humans, the trend appears to be similar. Only mild platelet dysfunction was observed in patients transfused with HBOC-201 [99,100], with no apparent effects on human platelet activation or function [101]. Clinically tested HBOCs do not produce relevant coagulopathy-associated syndrome [102]. The presence of HBOCs in circulation may, however, affect the pro- vs. anti-coagulation balance independently of NO, for example, through the dilution of the involved proteins, the presence of Hb^+^ that inhibits platelet aggregation, and the enhanced elimination of the von Willebrand factor [103].

### 3.4. Oxidative Stress

NO is itself a free radical that should be represented as NO●. Especially via the strong pro-oxidant ONOO^−^, which is not itself a free radical but originates from toxic superoxide and hydroxyl radicals [104], NO amplifies the oxidative injury carried out by reactive O_2_ species (ROS). Indeed, NO_2_^−^ infusion with HBOCs is profoundly cytotoxic in the lungs of a swine animal model [105], and NO_2_^−^ is known to accelerate Hb oxidation and induce tissue toxicity [106]. The major path underlying HBOCs’ cytotoxicity is the formation of reactive ferryl Hb derivatives that are subsequently reduced by NO to Hb^+^ and NO_2_^−^ [20]. The ferryl forms of Hb have been shown to trigger mitochondrial dysfunction and injury in alveolar type I cells [107]. The scavenging of vascular NO is thought to be the major cause of toxicity. However, based on more recent preclinical studies, oxidative pathways driven by the heme prosthetic group seem to play a more prominent role in the overall toxicity of free Hb or HBOCs [74]. Remarkably, by stabilizing the ferryl iron and the globin radical on βTyr145, haptoglobin binding to Hb dimers acts as an anti-oxidant feature [108].

Although the in vivo level of NO in circulation is normally below the critical concentration above which NO becomes injurious, the NO-scavenging effect might appear favorable because it removes the substrate needed to fuel the ROS chain. However, most experimental and observational evidence converges in highlighting that HBOC infusion exacerbates oxidative stress. For example, the degree of apoptosis in neurons, a hallmark of neuronal damage in an organ that is highly vulnerable to ROS [109,110] because of its high O_2_ usage and relatively under-expressed antioxidant defense [111], increases sharply in the presence of HBOCs [80]. One reason for this apparently contradictory behavior resides in the fact that HBOC infusion is predicted to increase the O_2_ supply to hypoxic or ischemic tissue abruptly, which drives reoxygenation or reperfusion injury, especially by means of a reaction catalyzed by xanthine oxidase that converts the ATP degradation product hypoxanthine into xanthine and urate with the production of superoxide anions, which feed the formation of hydrogen peroxide and hydroxyl radicals [112]. Most tissues possess an extraordinary ability to buffer pro-oxidant factors by building endogenous antioxidant defenses, such as the nuclear factor erythroid 2-related factor 2 (Nrf2) [113,114]. Also, protein kinase B or serine/threonine-specific protein kinase (Akt) plays a key role in neuroprotection [115,116], and reveals protective features that override those elicited by Nrf2 [117]. Akt phosphorylation and activation, thus, emerge as a central feature in the protection, elicited by NO, of either endogenous or exogenous sources in most tissues [118,119]. Therefore, NO scavenging by HBOCs is predicted to shift the pro- vs. antioxidant balance toward the formation of oxidative potential, as observed in almost all studies. Remarkably, a similar outcome occurs in hyperoxia-like situations (i.e., higher than normal PO_2_) where the development of antioxidant defenses may efficiently contrast the increased ROS production, thereby attenuating the onset of oxidative injury [117].

### 3.5. O_2_ Carrying Function

The main function of Hb is to carry O_2_ from the lungs to the tissues. This function is exploited by unique characteristics that can be summarized as follows: in the lungs, Hb must occur at a high affinity for the O_2_ state to favor O_2_ binding, but in the tissues the Hb affinity for O_2_ must be less to facilitate its release. This compromise is attained in intact RBCs by the particular form of the Hb-O_2_ dissociation curve that enables a high affinity for O_2_ at PO_2_ > 90–95 mmHg, i.e., the alveolar PO_2_, but moderate affinity at the PO_2_ values common in the capillary compartment, where O_2_ must be released [120]. The Hb-O_2_ affinity is marked by the P50 value, i.e., the PO_2_ at which half of the Hb is oxygenated, which is 26–28 mmHg in humans. As a matter of fact, most of the modified human Hbs often display relatively low P50 values in the range of 5-to-15 mmHg, which is considered a negative feature as it is believed that Hb has an intact capacity to bind O_2_ in the lungs but a low ability to release it to the tissues. This feature may be in part corrected by the analogs of 2,3-diphosphoglycerate (DPG), an effector of the Hb quaternary conformation that favors the low-affinity state. Several emulators of DPG have been evaluated such as pyridoxal phosphate [121]. Alternatively, acellular bovine Hb, which has a low affinity for O_2_ even outside the RBC because it binds chloride ions instead of DPG, is often considered as a valid substitute for human Hb [122]. In an alternative view of the O_2_ release system in the capillary compartment, it was pointed out that a high-O_2_-affinity Hb may even be beneficial in the presence of HBOCs because it avoids tissue oxygenation in the arteriole portion of the capillary, thereby preventing vasoconstriction in the remaining portion of the capillary [70].

In the RBC, the slow but continuous oxidation of Hb to Hb^+^, a form that does not have any O_2_-carrying property, is largely compensated by the enzyme Hb^+^ reductase, which critically depends on a continuous supply of NADPH through the reaction of glucose-6-phosphate dehydrogenase. In the presence of acellular Hb in plasma that does not have Hb^+^ reductase activity, this capacity to reconvert Hb^+^ to Hb is lost. Thus, Hb autoxidation represents an important challenge in HBOC development not simply for the loss of the O_2_ carrying capacity, but especially because of the increased release of heme and iron from Hb, which enhance the pro-oxidant potential in HBOC-transfused organisms through the Fenton reaction, which produces potentially injurious ROS. As a matter of fact, the modification of the β93Cys, e.g., the conjugation with maleimide-activated PEG, enhances the rate of Hb autooxidation and heme loss [123]. Many of the negative outcomes, such as iron overload and oxidative stress with damage to the kidney and the brain, are attributable to this chain of events that is triggered by acellular Hb.

## 4. Protein Engineering

Various protein-engineering options have been examined from time to time to overcome the limitations to the use of HBOCs in clinics. As stated above, the purpose here is not to review the various options, but only to identify the link between them and the issues related to NO scavenging.

### 4.1. Stabilizing the Hb Tetramers

The splitting of Hb tetramers into dimers represented perhaps the first challenge in the search for a suitable HBOCs. Since the early studies performed to determine the mechanism of Hb binding with ligands, it has become evident that the absolute concentration of Hb represents an important contributor to tetrameric Hb dimerization [124]. The reaction of a Hb-tetramer to 2 Hb-dimers is a disproportionation reaction, which implies that the relative abundance of the reactants depends on their absolute concentration. As the K_eq_ of the dimerization reaction is about 10^−6^ M [125], this implies that a tenfold protein dilution in the mM range, i.e., the typical Hb concentration in the RBC, nearly doubles the amount of Hb dimers [126]. When HBOC are infused into circulation, the Hb concentration in the plasma is much lower than in the RBC, which forms considerable amounts of Hb dimers that do not display allosteric effects, lack the characteristic sigmoid shape of the HbO_2_ dissociation curve, do not react with DPG, increase the oncotic colloid pressure and, with MW = 32 kDa, tend to extravasate faster in the interstitial space and be lost in the kidney filtrate.

The dimerization of Hb is required for the reaction with haptoglobin, a plasma protein with a high affinity for Hb dimers that targets Hb for clearance without damage to the kidneys and with minimal lipid peroxidation and inflammation [127]. Accordingly with the properties of Hb dimers described above, the heme active sites of the Hb subunits bound to haptoglobin display higher reactivity with O_2_ and NO_2_^−^ with respect not only to the Hb tetramers, but also to the free unbound Hb dimers, because of the enhanced access of the ligands and a decrease in the redox potential in haptoglobin-bound Hb dimers [128]. This translates into major effectiveness of the NO regeneration function of the Hb-haptoglobin complexes, with particular benefit for the control of inflammation. In addition, haptoglobin binding stabilizes Hb ferryl iron and decreases the free-radical reactivity of Hb, which translates into improved protection against Hb-induced damage to the vasculature [108]. In agreement with this view, the sequestration of extracellular Hb within a complex with haptoglobin, rather than the modulation of its NO- and O_2_-binding characteristics, has been shown to be efficient in decreasing its hypertensive and oxidative effects in dogs and guinea pigs [129].

To overcome the problem associated with tetramer Hb splitting into dimers, several protein engineering tools have provided methods to cross-link the Hb dimers with covalent bonds within the Hb molecule, thereby preventing its dimerization. One example is HbXL99a, a human Hb where the α chains are cross-linked with bis(3,5-dibromosalicyl) fumarate between two Lys residues [130]. Another example is represented by diaspirin cross-linked Hb, obtained by reacting deoxygenated Hb with bis(3,5-dibromosalicyl) fumarate to form a stabilized tetramer that is covalently linked between the α globin chains [131,132]. Despite providing favorable chemical–physical characteristics with reasonable side effects in preclinical models [133,134], a large-scale randomized clinical trial in 112 patients with severe, uncompensated, traumatic hemorrhagic shock showed that the mortality was higher for patients in the diaspirin cross-linked Hb group [135], for reasons ascribable to the NO-scavenging effect.

At present, studies are in progress to design new, more flexible reagents to cross-link and connect tetramers either with angular connectors that permit torsional movement, or with linear connectors that resemble previously studied systems. The resulting cross-linked tetramers were produced in a high yield and were isolated and characterized [136], but we are not aware of successful clinical trials employing these derivatives. Remarkably, YQ23, a bovine-derived cross-linked Hb, more resistant to oxidation, with a low Hb^+^ content (<4.8%) and P50 = 40 mmHg, was tested in rats and pigs, with favorable outcomes in terms of hypotensive resuscitation, stabilized hemodynamics, an increased tissue O_2_ consumption, increased cardiac output, and reduced liver and kidney injury [137], but we are not aware of any clinical trials performed with this HBOC.

### 4.2. Hb Polymerization

In addition to the intramolecular stabilization of Hb tetramers, several trials have addressed intermolecular polymerization procedures to increase the apparent MW, while retaining the advantages conveyed by intramolecular cross-linking through diacid [138] or glutaraldehyde [139] cross-linking. Polymerized Hb revealed satisfactory tolerance in clinical trials aimed at supporting prehospital ambulance patients in hemorrhagic shock despite an increase in myocardial infarction events [140], a possible side effect of persisting NO scavenging. Due to the oxidative stress linked to resuscitation and cardiovascular events, polymerized Hb solutions with catalase/superoxide dismutase activity were designed [141], with favorable outcomes in a rat model of transient global brain ischemia–reperfusion [142]. Although the association of polymerized Hb with antioxidants has become an emerging relevant feature in HBOC development [143], clinical trials developed using these products, summarized in [5], have still displayed critical issues related to vasoactivity and hypertension [144]. Indeed, the issues related to NO are not covered by the interventions aimed solely at modifying the size of the Hb molecule and the antioxidant capacity. The MW of infused Hb molecules, however, remains a pivotal determinant of the outcome. For example, its presence in a solution containing high-MW polymerized bovine Hb of 15% polymers with MW < 256 kDa caused a marked elevation in mean arterial pressure accompanied by changes in markers related to systemic inflammation and kidney, liver, and lung injury [145].

### 4.3. Hb Conjugation

The PEGylation of Hb provides the benefit of increasing the molecular radius of the Hb molecule to prevent filtration in the renal glomeruli and in vascular beds, with a consequent longer persistence in circulation and less tissue injury. A variety of chemical strategies for attaching PEG to Hb has resulted in different products and options [146]. Of the several PEGylated Hbs proposed so far, a maleimide-activated PEG-conjugated human Hb [147] has an average of six PEG chains (MW 5 kDa each) per Hb tetramer, which creates a shell of water surrounding the protein. While maleimide-based PEGylation does not modify the native structure of adult Hb-O_2_, in deoxygenated Hb it loosens the dimer-to-dimer contacts, thereby favoring dimerization [148]. Decorating the surface of Hb increases the Hb-O_2_ affinity and decreases cooperativity [17,149]. Incidentally, PEGylated bovine Hb shows a higher hydrodynamic volume, colloidal osmotic pressure, and viscosity than human adult Hb, but with the advantage that the high P50 of bovine Hb offsets the PEGylation-induced perturbation in the conformation of human Hb [150]. As far as the NO-scavenging effect is concerned, human PEGylated Hb exhibits similar kinetic characteristics of the reaction with gasses as adult human Hb [18]. Yet, this PEGylated Hb appears not to elicit the hypertensive response typically observed in most other acellular Hbs [17], in agreement with independent observations in a conscious-hamster dorsal-skinfold model that documented lesser vasoconstriction and hypertension upon the infusion of larger HBOCs in the 7–224 nm diameter range [151]. Together, such results address the mechanisms underlying the relationship between vasoactivity and the NO-scavenging effect. More recently, a new hyperpolymerized human Hb was developed with several novel characteristics that include a high average MW (1.6 MDa), high Hb concentration (11 g/dL), low colloid osmotic pressure (12 mmHg), and high viscosity (12 cPs) [152]. Remarkably, as with PEGylated Hb, this product did not exhibit adverse effects such as localized vasoconstriction or hypertension [152].

More recently, new reactive thiol residues were engineered at sites distant to the heme group and the dimer/dimer interface, e.g., βCys93Ala/αAla19Cys and βCys93Ala/βAla13Cys, with no alterations in the affinity nor cooperativity of Hb-O_2_, but decreased rates of autoxidation and heme release [153]. No direct data on NO reactivity, however, are available.

### 4.4. Recombinant Hb

The availability of site-directed mutagenesis techniques has enabled the production of recombinant human Hb variants targeted at specific functions. Today, engineered *Saccharomyces cerevisiae* or *Corynebacterium glutamicum* strains can produce up to 18–20% of total cell proteins as human Hb [154,155] at reasonable costs and high yields: a single 100 m^3^ tank may produce 200 kg of Hb, i.e., the equivalent of 1300 L of blood [156]. Observing the characteristics of naturally occurring human Hb mutations provides lessons on how to design recombinant Hb [157]. Among the several examples related to HBOCs [158], Hb Providence, with a low O_2_ affinity [159] and enhanced stability in pro-oxidant media [160], has emerged to address the β82Lys→Asn mutation functional to confer oxidative stability to Hb [161]. By contrast, fetal Hb did not display significant advantages with respect to adult Hb in the properties related to NO and NO_2_^−^ reductase reactivity [162], despite improved oxidative stability [163]. However, surface modifications can suppress Hb recognition by haptoglobin, immunoglobulins, and anti-Hb antibodies [164].

As β93Cys plays a key role in NO binding, O_2_-dependent conformational changes, and cooperativity, this solvent-accessible residue has become a gateway to modulate the NO-scavenging activity and oxidative stability of HBOCs [165]. Hb Minotaur contains α-human and β-bovine chains [166]. The polymerized form of this Hb, with NO-reactive β93Cys residues replaced by non-reactive residues, revealed favorable characteristics in its plasma retention time and vasoactivity, and remarkably reduced ischemic damage in the brain despite similar P50 as adult Hb [167]. In another trial, the β93Cys→Gly mutation, followed by tetramer S-S stabilization, produced a stable octameric Hb with conserved gas-binding kinetic features [168]. However, despite promising features, no FDA license was granted to recombinant Hb products for use in humans until 2013 [169], and we are not aware of any license given after this date.

### 4.5. Polynitroxylated Hb

In polynitroxylated Hb, the addition of nitroxyl groups, whose chemistry implies the release of NO [170], to PEGylated Hb combines the advantages of large Hb tetramers with those led by the cryoprotection associated with NO [171]. Manufactured as ∼8 nm nanoparticles with bovine Hb as the protein center, polynitroxylated Hb was successfully tested in contexts related to resuscitation after traumatic brain injury and/or hemorrhagic shock [172,173,174]. NO release from this compound helped maintain the dilation of pial arteries, thereby improving collateral blood flow, and reducing infarct volume in ischemic stroke [175]. Remarkably, polynitroxylated Hb can be successfully stored as HbCO, which enables two distinct purposes. First, the robust heme-CO binding prevents Hb autooxidation, which extends its shelf life. Second, when infused into the circulation, the slow CO release and re-equilibration with RBC Hb exert anti-inflammatory effects, as observed with other CO donors [176,177]. Indeed, upon the infusion of polynitroxylated HbCO in traumatic brain injury with hemorrhagic-shock pigs, HbCO disappeared within one day, presumably favoring distribution of this gas to tissues, with the alleviation of injury to the neocortical gray and white matter as indicated by maintained amyloid precursor protein, dendrite length and microglial activation levels [178].

The NO-containing moieties conjugated to Lys residues in polynitroxylated Hb also contribute to protect the endothelium from oxidative stress and leukocyte adhesion. Indeed, the reaction of nitroxyl moieties with superoxide anions [179,180] inhibits peroxide and superoxide-mediated neutrophil adherence to endothelial cells [181], free radical-induced microcirculatory dysfunction [91], and the oxidation of hydrogen peroxide coupled with nitroxide (=NO•) [182]. It was shown that the superoxide dismutase/catalase effect in polynitroxylated Hb offers additional neuroprotection in the resuscitation of guinea pigs after hemorrhagic shock combined with traumatic brain injury [183].

### 4.6. Microencapsulation

The first published report of microcapsules constituted by semi-permeable barriers to emulate RBCs dates back to 1964 [138]. Capsules are made of a variety of substances that include synthetic polymers, negatively charged polymers, crosslinked proteins, lipid–proteins, lipid–polymers, surface polysaccharides, and others [184]. Technical advancements have enabled the successful replacement of some of the circulating RBCs in rats, to be effective in hemorrhagic shock [185,186]. In some formulations, the lipid bilayers of the nano-artificial RBCs are permeable to O_2_, CO, glucose, and lactate [184]. Although there are little data related to the NO passage across artificial lipid bilayers, some points are, nevertheless, established. The encapsulation of concentrated Hb in phospholipid vesicles with diameters from 50 to 8000 nm retards the reaction of Hb with NO, but not with CO, due to resistance attributed to the intracellular diffusion barrier [187]. Such resistance may be associated with the transformation of gaseous NO into inorganic salts, as shown in Figure 2, while CO reacts with Hb only as a gas. The bimolecular rate constant for the Hb + NO binding decreases with an increasing intracellular Hb concentration and with the vesicle diameter, potentially indicating that increasing the viscosities of the Hb solutions and the increased diffusional distances of NO inside the vesicles of larger diameters may be the true limiting factors for NO scavenging [187]. This finding, however, was only in part confirmed by another study that highlighted that the major resistance to NO scavenging by microencapsulated Hb is to be attributed to the membrane permeability of NO (that lies in the 10^5^–10^3^ μm s^−1^ range), depending on the formulation of the lipid bilayer, and secondarily to the extracellular diffusion of NO (3300 μm^2^ s^−1^) to the surface of the microparticle [32]. By contrast, in intact RBCs, the extracellular diffusion of NO to the RBC membrane surface constitutes the major limiting factor. Remarkably, in the oxygenated state the RBC membrane is 12.5 times less permeable to NO than in the deoxygenated state, as a consequence of deoxygenated Hb binding to Band 3, or anion exchanger-1, in the cytoskeleton [188,189].

Although the issues related to NO scavenging by microencapsulated Hb may appear still controversial, this technique enables the incorporation of other proteins with various roles into the macrovesicles, for example carbonic anhydrase to enhance carbon dioxide transport and catalase/superoxide dismutase to enhance the antioxidant function [190]. This formulation proved efficient in a rat hemorrhagic-shock model with a two-thirds blood-volume loss and 90 min of sustained shock, to enable faster recovery of the heart from ischemia, better histological findings in the post-ischemic intestine, and more favorable tests for anaphylactic reactions [191].

### 4.7. Earthworms and Marine Worm Erythrocruorin

In earthworms (*Lumbricus terrestris*), the lack of RBCs, the replacement of mammal Hb with erythrocruorin, a large 3.6 MDa MW [192] multiprotein complex with 144 heme-containing globins [193] provide a unique opportunity to develop alternative HBOCs. Other advantages include the P50, which is similar to that of human blood, and its endogenous superoxide dismutase activity that protects from ROS [192]. The properties of erythrocruorin have been reviewed recently [194]. Extremely stable and resistant to oxidation, erythrocruorin occurs also in *Arenicola marina*, also known as marine worm, and has been tested as an efficient O_2_ carrier with potential anti-inflammatory, anti-bacterial, and antioxidant properties [195]. Remarkably, erythrocruorin displays a reduced NO-scavenging capacity with consequent blunted vasoconstriction and hypertension [196,197], perhaps due to molecular characteristics that block NO dioxygenation [198]. Its NO-binding rate with erythrocruorin, lower than that of adult human Hb [199], enables erythrocruorin administration in rodents with only a transient non-significant increase in blood pressure with respect to saline [200]. Erythrocruorin possesses NO_2_^−^ anhydrase activity that leads to the formation of N_2_O_3_ bound to the heme in HbFe^2+^ [199]. As the rate of formation of this proposed intermediate was slower than the rate of nitrosylation, it is possible that under physiological conditions with low NO levels, erythrocruorin may not be effective as HBOCs as some PEGylated Hbs [201]. The in vivo data, however, indicate that erythrocruorin is effective concerning O_2_ delivery under conditions where NO is not depleted.

Easily scalable and affordable techniques such as tangential-flow filtration enable one to produce highly purified erythrocruorin from earthworms with a yield of 5–10 g protein/1000 worms [197]. In clinical contexts, such products were shown to be effective to improve renal function in kidney transplant recipients [202], to attenuate ischemia–reperfusion injury during the cold storage of steatotic livers [203], to prolong pulmonary preservation during transplant [204], to improve liver-graft preservation [205], and to decrease bacterial inflammation [206] and the injury led by hypoxia in periodontal healing [207].

## 5. Conclusions

The development of HBOCs is continuing >50 years after its first trials. Many bottlenecks have occurred. While most of them have been solved, or are being solved by various protein engineering approaches, the fast binding of NO with Hb appears one of the barriers that are considerably hindered the launch of these life-saving therapeutics. A basic understanding of the phenomena underlying the vasoconstrictive effect led by plasma Hb, namely HbSNO [13] and nitrite reductase activity [14], are key to focus HBOC research on reliable solutions stringently needed in the modern society.

## Figures and Tables

**Figure 1 ijms-24-14902-f001:**
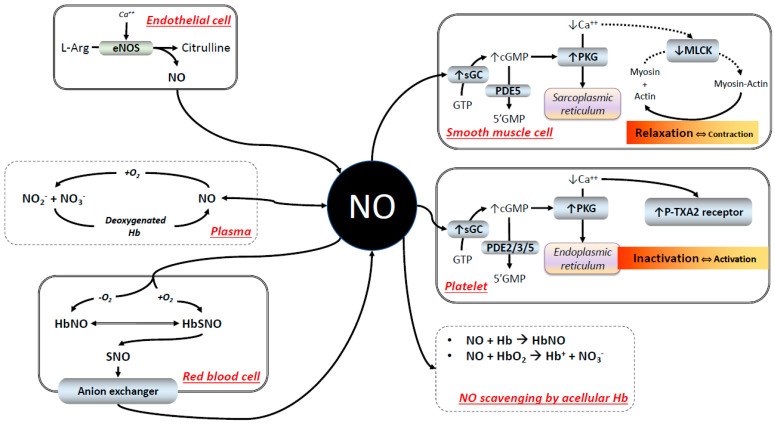
Simplified scheme that summarizes the pleiotropic roles of NO in circulation and the main factors that regulate its bioactivity. Explanation and abbreviation meanings in the text. The symbols ↓ and ↑ highlight decrease and increase, respectively.

**Figure 2 ijms-24-14902-f002:**
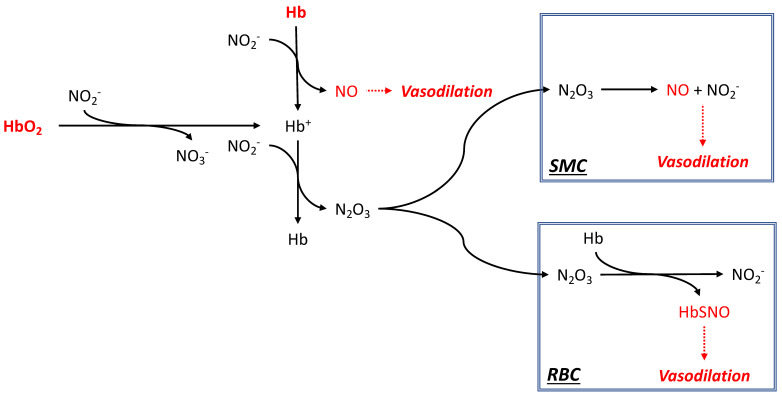
Simplified putative scheme illustrating the nitrite reductase activity exerted by Hb in either the deoxygenated (Hb) or oxygenated (HbO_2_) form. Nitrite (NO_2_^−^) reduction by Hb forms NO and oxidized methemoglobin (Hb^+^). The same end product is eventually formed from HbO_2_ with the concomitant oxidation of NO_2_^−^ to nitrate (NO_3_^−^), along with a series of free radicals and intermediates [72]. The reconversion of Hb^+^ to Hb forms N_2_O_3_, a stable diffusible form that is taken up by smooth muscle (SMC) and red blood (RBC) cells. While in SMC N_2_O_3_ reforms NO and NO_2_^−^, in the RBC N_2_O_3_ nitrosylates thiols to form nitrosylated Hb (HbSNO) [73]. The stoichiometric balance of the various reactions is not respected for clarity.

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
