# Peer review of "How Nitric Oxide Hindered the Search for Hemoglobin-Based Oxygen Carriers as Human Blood Substitutes"

_ijms, 2023, doi:10.3390/ijms241914902_

Round 1
Reviewer 1 Report
This manuscript is well written and certainly deserves to be published. It is a highly relevant topic to be included in this volume. I have a few minor points which the authors may consider:
1. The section covering pegylation appears very dated. A lot of work has been published in this field, especially on site-directed pegylation of Hb.
2. The same holds true for the recombinant production of Hb. Several recent studies have been omitted and the yields are now much higher than stated in the manuscript.
3. I would also lika to see some comments on haptoglobin. Can Hp-Hb complexes influence NO chemistry? If so, in which ways?
Author Response
Reviewer 1 - Round 1
Thank you for the time and energy spent reviewing this article.
This manuscript is well written and certainly deserves to be published. It is a highly relevant topic to be included in this volume. I have a few minor points which the authors may consider:
- The section covering pegylation appears very dated. A lot of work has been published in this field, especially on site-directed pegylation of Hb.
We agree with the Reviewer, but note that, as outlined in the last sentences of the Introduction “…the purpose of this article in the special issue dedicated to the 25th anniversary of NO is not to add a further review of past and present HBOC candidates but rather to focus on the basic biochemical and physiological events that hindered the launch of a single HBOC candidate in the clinical arena, with special concern to one of the several interactions of Hb with other molecules: the reaction with NO.” After a deep review of available information in the scientific literature, excluding abstracts, patents, book chapters, and other non-reviewed reports, we did not find much concerning the interaction of Hb with NO. However, subchapter 4.3 Hb conjugation was rewritten entirely in the light of available information.
The PEGylation of Hb provides the benefit of increasing the molecular radius of the Hb molecule to prevent filtration in the renal glomeruli and in vascular beds, with consequent longer persistence in the circulation and less tissue injury. A variety of chemical strategies for attaching PEG to Hb resulted in different products and options [141]. Of the several PEGylated Hbs proposed so far, a maleimide-activated PEG-conjugated human Hb [142] has an average of six PEG chains (MW 5 kDa each) per Hb tetramer, which creates a shell of water surrounding the protein. While maleimide-based PEGylation does not modify the native structure of adult Hb-O2, in deoxygenated Hb it loosens the dimer-to-dimer contacts, thereby favoring dimerization [143]. Decorating the surface of Hb increases the Hb-O2 affinity and decreases cooperativity [16, 144]. Incidentally, PEGylated bovine Hb shows higher hydrodynamic volume, colloidal osmotic pressure, and viscosity than human adult Hb, but with the advantage that the high P50 of bovine Hb offsets the PEGylation-induced perturbation in the conformation of human Hb, still resulting non-vasoactive [145]. As far as the NO scavenging effect is concerned, human PEGylated Hb exhibits similar kinetic characteristics of the reaction with gasses as adult human Hb [17]. Yet, this PEGylated Hb appears not to elicit the hypertensive response typically observed in most other acellular Hbs [16], in agreement with independent observations in conscious hamster dorsal skinfold model that documented lesser vasoconstriction and hypertension upon infusion of larger HBOC in the 7-224 nm diameter range [146]. Together, such results address the mechanisms underlying the relationship between vasoactivity and the NO scavenging effect. More recently, a new hyperpolymerized human Hb was developed with several novel characteristics that include a high average MW (1.6 MDa), high Hb concentration (11 g/dL), low colloid osmotic pressure (12 mmHg), and high viscosity (12 cPs) [147]. Remarkably, as with PEGylated Hb, this product did not exhibit adverse effects such as localized vasoconstriction or hypertension [147].
More recently, new reactive thiol residues were engineered at sites distant to the heme group and the dimer/dimer interface, e.g., βCys93Ala/αAla19Cys and βCys93Ala/βAla13Cys, with no alterations of the Hb-O2affinity nor cooperativity, but decreased rates of autoxidation and heme release [148]. No direct data on NO reactivity, however, is available.
- The same holds true for the recombinant production of Hb. Several recent studies have been omitted and the yields are now much higher than stated in the manuscript.
Even the subchapter 4.4 Recombinant Hb has been completely rewritten to accommodate the Reviewer’s criticism. As outlined above, please consider that the aim of this article was not to review the advances in HBOC studies, as already egregiously performed in several other articles, but to focus on the NO-related aspects of HBOC.
The availability of site-directed mutagenesis techniques enabled the production of recombinant human Hb variants targeted at specific functions. Today, engineered Saccharomyces cerevisiae or Corynebacterium glutamicum strains can produce up to 18-20% of total cell proteins as human Hb [149, 150] at reasonable costs and high yields: a single 100 m3 tank may produce 200 kg of Hb, i.e., the equivalent of 1300 L of blood [151]. Observing the characteristics of naturally occurring human Hb mutations provides lessons on how to design recombinant Hb [152]. Among the several examples related to HBOC [153], Hb Providence, with low O2 affinity [154] and enhanced stability in pro-oxidant media [155], emerged to address the β82LysAsn mutation functional to confer oxidative stability to Hb [156]. By contrast, fetal Hb did not display significant advantages with respect to adult Hb in the properties related to NO and NO2- reductase reactivity [157], despite improved oxidative stability [158]. However, surface modifications can suppress Hb recognition by haptoglobin, immunoglobulins, and anti-Hb antibodies [159].
As β93Cys plays a key role in NO binding, O2-dependent conformational changes, and cooperativity, this solvent-accessible residue became a gateway to modulate the NO scavenging activity and oxidative stability of HBOC [160]. Hb Minotaur contains α-human and β-bovine chains [161]. The polymerized form of this Hb, with NO-reactive β93Cys residues replaced by non-reactive residues, revealed favorable characteristics in plasma retention time and vasoactivity, and remarkably reduced ischemic damage in the brain despite similar P50 as adult Hb [162]. In another trial, the β93CysGly mutation, followed by tetramer S-S stabilization, produced a stable octameric Hb with conserved gas-binding kinetic features [163]. However, despite promising features, no FDA license was granted to recombinant Hb products for use in humans until 2013 [164], and we are not aware of any license given after this date.
- I would also lika to see some comments on haptoglobin. Can Hp-Hb complexes influence NO chemistry? If so, in which ways?
We thank the Reviewer for this suggestion and have added a new paragraph in subchapter 4.1 Stabilizing the Hb tetramers to encompass this important issue.
The dimerization of Hb is required for the reaction with haptoglobin, a plasma protein with a high affinity for the Hb dimers that targets Hb to clearance without damage to kidneys and with minimal lipid peroxidation and inflammation [122]. Accordingly with the properties of Hb dimers described above, the heme active sites of the Hb subunits bound to haptoglobin display higher reactivity with O2 and NO2- with respect not only to the Hb tetramers, but also to the free unbound Hb dimers, because of an enhanced access of the ligands and a decrease in the redox potential in haptoglobin-bound Hb dimers [123]. This translates into major effectiveness of the NO regeneration function of the Hb-haptoglobin complexes, with particular benefit for the control of inflammation. In addition, haptoglobin binding stabilizes Hb ferryl iron and decreases free radical reactivity of Hb, which translates into improved protection against Hb-induced damage to the vasculature [103]. In agreement with this view, sequestration of extracellular Hb within a complex with haptoglobin, rather than modulation of its NO- and O2-binding characteristics, has been shown to be efficient in decreasing its hypertensive and oxidative effects in dogs and guinea pigs [124].

Reviewer 2 Report
This is a wide ranging and generally useful review of how the physiological roles of NO are impacted by introducing HBOC’s into the blood stream. It provides an extensive list of references that help interested readers navigate the literature on this important area. However, the manuscript requires amendment before publication. The reasons for this for this are:
aa The text is unclear and clumsy in several places making it difficult to understand.
bb There are sections in which one view of the field is given, ignoring counter evidence and views, these should be added.
cc There are places where word selection is inappropriate, again leading to misunderstanding,
I list some places where attention is required and also suggest the authors read their manuscript again to ensure that it is clearly understandable.
.
Title, and in the text (line 518) , “harshened” is a strange word to use, perhaps “impeded/impedes” is better
Line 40, better “As NO has a higher affinity for Hb than O2 (even higher than CO) this interferes with its O2 carrying function, making NO a toxic gas”
Lines 45 -50 is one sentence. The authors meaning would be clearer if this were broken into shorter sentences.
Line 50-63. See also below
Line 59 caliber is usually reserved for gun barrels, how about diameter?
Line 67 or should be for
Line 83, The extremely rapid dioxygenation reaction between oxyHb and NO forming nitrate should be more clearly explained, as it is the major route the Hb scavenges NO. See the work of of Herold and others.
Lines 90-103. Regarding the beta93cys – SNOHb hypothesis, the counter view is missing. I suggest the authors give a more balanced perspective. The authors may refer to papers showing that there are no differences in hypoxic blood flow and vasodilation effects ex vivo and in vivo between WT and b93A mice (Sun CW et al Circulation. 2019; Isbell et al Nat Med 2008). Also, the potential that cardiac phenotypes observed in b93A mice may be due to the conserved b93cys playing a role in redox homeostasis mechanisms in the red cells need to be evaluated as an alternate explanation to any SNO-dependent effects (Vitturi et al 2013 FRBM). Finally, SNOHb has a higher oxygen affinity than non-SNOHb which poses a significant challenge since SNOHb has to be deoxygenated for proposed hypoxic SNO-transfer pathway from b93ys (Patel et al JBC 1999)…..so S-nitrosation of b93cys is probably not a good idea just from oxygen offloading perspective.
Line 132 delete “the” before NO
Line 134 delete “the” before most
Line 138 refute is more usual than the rare confute.
Line 143 The statement in this sentence is not universally accepted see the work of R.Patel (Alabama).
Line 178 “points” is better than “conveys”.
Line264-267 unclear and rather long sentence.
Line 268 pararaph. Oxidative stress is discussed but no attention is paid to the extensive work showing that haemoglobin reacts with peroxide to form ferryl species and that this is responsible for pathological consequences, not just free iron. Alayash writes extensively about this and argues that heme induced oxidative stress is as important as NO in HBOC toxicity.
Line 269 “represented” rather than mentioned.
Line 288 reveals.
Line 293 what does analog mean here?
Line 300 I don’t think egregiously is the right word here.
Line 310 insert “such” before as pyridoxal
Line 339 better “the absolute concentration of Hb is an important”.
Line346 forcefully? Better necessarily.
Line 347 “addressed” better than approached, “while” better than yet.
Line 402 “elicit” better than exhibit.
Line 412 “has been” better than was.
Line 428/429 “of” better than led by.
Line 445 delete groups.
Line 462 difficult to understand the effect of encapsulation on NO but not CO. Is the NO present not as the gas but as one of nitrogen compounds such as mentioned in Fig 2?
The English needs attention in places and I have offered suggestions in the report above.
Author Response
Reviewer 2- Round 1
Thank you for the time and energy spent reviewing this article. We are confident that they greatly helped in improving this manuscript.
This is a wide ranging and generally useful review of how the physiological roles of NO are impacted by introducing HBOC’s into the blood stream. It provides an extensive list of references that help interested readers navigate the literature on this important area. However, the manuscript requires amendment before publication. The reasons for this for this are:
- The text is unclear and clumsy in several places making it difficult to understand.
- There are sections in which one view of the field is given, ignoring counter evidence and views, these should be added.
- There are places where word selection is inappropriate, again leading to misunderstanding,
I list some places where attention is required and also suggest the authors read their manuscript again to ensure that it is clearly understandable.
Thank you for all the very useful suggestions!
Title, and in the text (line 518) , “harshened” is a strange word to use, perhaps “impeded/impedes” is better
After extensive discussion, we opted to use the term “hinder”:
All three words refer to the action of making something difficult or impossible to do. However, they can have slightly different connotations.
To hinder means to prevent progress or make something difficult to do. For example, "His injury hindered his ability to play sports."
To hamper also means to impede progress or make something difficult to do, but it often implies that something is being physically blocked or restricted. For example, "The heavy snow hampered our efforts to clear the driveway."
To impede means to slow down or block progress. It often implies that something is being deliberately blocked or hindered. For example, "The new regulations impeded the company's growth."
Line 40, better “As NO has a higher affinity for Hb than O2 (even higher than CO) this interferes with its O2 carrying function, making NO a toxic gas”
We rephrased that sentence as “Having a higher affinity for Hb than O2, even higher than carbon monoxide (CO), NO appears as a toxic gas because it negatively interferes with the Hb’s O2 carrying function”.
Lines 45 -50 is one sentence. The authors meaning would be clearer if this were broken into shorter sentences.
Done, that too-long sentence is now split into three shorter ones.
Line 50-63. See also below
See below.
Line 59 caliber is usually reserved for gun barrels, how about diameter?
Done.
Line 67 or should be for
Done.
Line 83, The extremely rapid dioxygenation reaction between oxyHb and NO forming nitrate should be more clearly explained, as it is the major route the Hb scavenges NO. See the work of of Herold and others.
The Reviewer is right in commenting that we did not explain in detail this critical reaction and underestimated the work by S. Herold. In the revised version, we added a few sentences (subchapter 2.1) to fulfill this failure.
The bimolecular reaction rate of 20 μM−1 s−1 for the NO-driven Hb oxidation is however of the same order of magnitude as the bimolecular association of NO to deoxygenated Hb [17]. This reaction is more complex than it appears because of the formation of a peroxynitrito-complex Hb+OONO that decays rapidly to Hb+ and NO3- [18]. The mechanism is further complicated by the fact that, unlike O2 and CO, NO can bind Hb+ forming an unstable HbNO+ complex. Severe pro-oxidant conditions may generate the reactive ferrylHb derivative that is reduced by NO to Hb+ and NO2- [19].
Lines 90-103. Regarding the beta93cys – SNOHb hypothesis, the counter view is missing. I suggest the authors give a more balanced perspective. The authors may refer to papers showing that there are no differences in hypoxic blood flow and vasodilation effects ex vivo and in vivo between WT and b93A mice (Sun CW et al Circulation. 2019; Isbell et al Nat Med 2008). Also, the potential that cardiac phenotypes observed in b93A mice may be due to the conserved b93cys playing a role in redox homeostasis mechanisms in the red cells need to be evaluated as an alternate explanation to any SNO-dependent effects (Vitturi et al 2013 FRBM). Finally, SNOHb has a higher oxygen affinity than non-SNOHb which poses a significant challenge since SNOHb has to be deoxygenated for proposed hypoxic SNO-transfer pathway from b93ys (Patel et al JBC 1999)…..so S-nitrosation of b93cys is probably not a good idea just from oxygen offloading perspective.
The Reviewer is perfectly right. We have completely rewritten two entire paragraphs under subchapter 2.1 to accommodate the counter view of the HbSNO hypothesis citing the suggested studies and others. The two rewritten paragraphs now read:
When intact RBCs are in an O2-rich environment such as arterial blood, the competition with O2 for the iron heme redirects NO to bind human Hb at a strictly conserved Cys residue in position β93 (β93Cys) to form the nitrosothiolated derivative HbSNO. The underlying molecular mechanism is supported by crystallographic studies showing that β93Cys is highly reactive in HbO2, but unreactive in deoxygenated Hb [20]. This feature enables considering Hb not simply as a NO scavenger, but as a transporter of NO to O2-poor tissues. In fact, increased fraction of deoxygenated Hb favors the intramolecular transfer of NO from β93Cys to the heme, with competitive binding of NO to RBC membrane band 3 anion exchanger that exports NO outside the RBC into the bloodstream to increase the flow of blood and O2 [12]. This mechanism may be particularly relevant in local hypoxia in the presence of greater fractions of deoxygenated Hb. The availability of knock-in humanized mice carrying the β93CysAla mutation might validate this hypothesis but gave controversial results. On one hand, the hypoxia-regulated HbSNO-mediated vasodilation was confirmed in vivo, despite numerous compensations in animals carrying the β93Ala mutation to alleviate tissue hypoxia [21]. On the other hand, no appreciable differences between β93Cys and β93Ala mice were detected in several models that included the resistance to myocardial ischemia/reperfusion, the export of NO bioactivity in human platelets co-incubated with RBCs, the vascular reactivity in aortas pre-treated with S-nitrosoCys, hemolysates or HbNO, as well as the hypoxic vasodilation in dorsal skin-fold windows [22]. Likewise, no differences in systemic and pulmonary hemodynamics were detected under hypoxia between the two groups of mice [23]. These observations apparently do not support the relevance of HbSNO-based mechanisms from the O2offloading perspective, despite greater ROS formation in β93Ala vs β93Cys RBCs and lungs, that nevertheless highlight a critical role for β93Cys in the antioxidant network [24].
Although this mechanism would be central to enable Hb to release NO depending on the hypoxia state, or the Hb-O2 saturation, there are arguments against this hypothesis based on both the stoichiometry of the Hb binding with NO and on the RBC environment. First, the formation of HbSNO in vitro was questioned due to technical artifacts attributed to the very fast binding of NO with HbO2 [25]. Second, similar rates of NO binding to HbO2 and Hb indicate independence on the Hb-O2 saturation [26, 27]. Third, the transnitrosation from HbSNO to the RBC membrane band 3 anion exchanger, which exports NO to the circulation, may be too slow to take place within the time spent by RBCs along the capillary [28]. Fourth, increased Hb-O2 affinity in HbSNO implies that Hb must flow through environments at very low PO2 in order to be deoxygenated enough to act as a NO transporter [28]. Thus, the role of Hb as a NO scavenger or NO transporter in vivo is at least controversial and accurate measurements of HbSNO stability in arterial and venous blood are required [29]. Also, the possible reaction of Hb+ with NO may perhaps be pivotal to understand the complex underlying mechanism [30].
Line 132 delete “the” before NO
Done.
Line 134 delete “the” before most
Done.
Line 138 refute is more usual than the rare confute.
Done.
Line 143 The statement in this sentence is not universally accepted see the work of R.Patel (Alabama).
Correct, we opted for removing that controversial sentence, which after all does not change appreciably the meaning of that paragraph.
Line 178 “points” is better than “conveys”.
Done.
Line264-267 unclear and rather long sentence.
We have rewritten that sentence, which we hope is now clearer:
The presence of HBOC in the circulation may however affect the pro- vs anti-coagulation balance independently of NO, as through the dilution of the involved proteins, the presence of Hb+ that inhibits platelet aggregation, and enhanced elimination of the von Willebrand factor [98].
Line 268 pararaph. Oxidative stress is discussed but no attention is paid to the extensive work showing that haemoglobin reacts with peroxide to form ferryl species and that this is responsible for pathological consequences, not just free iron. Alayash writes extensively about this and argues that heme induced oxidative stress is as important as NO in HBOC toxicity.
The Reviewer is right. The first paragraph of the sub-chapter 3.4 Oxidative stress was reworded to accommodate this lack of accuracy.
NO is itself a free radical that should be represented as NO●. Especially via the strong pro-oxidant ONOO-, which is not itself a free radical but originates toxic superoxide and hydroxyl radicals [99], NO amplifies the oxidative injury carried out by reactive O2 species (ROS). Indeed, NO2- infusion with HBOC is profoundly cytotoxic in the lungs of a swine animal model [100], and NO2- is known to accelerate Hb oxidation and induce tissue toxicity [101]. The major path underlying HBOC cytotoxicity is the formation of reactive ferryl Hb derivatives that are subsequently reduced by NO to Hb+ and NO2- [19]. The ferryl forms of Hb have been shown to trigger mitochondrial dysfunction and injury in alveolar type I cells [102]. Scavenging of vascular NO is thought to be the major cause of toxicity. However, based on more recent preclinical studies, oxidative pathways driven by the heme prosthetic group seem to play a more prominent role in the overall toxicity of free Hb or HBOCs [69]. Remarkably, by stabilizing the ferryl iron and the globin radical on βTyr145, haptoglobin binding to Hb dimers acts as an anti-oxidant feature [103].
Line 269 “represented” rather than mentioned.
Done.
Line 288 reveals.
Done.
Line 293 what does analog mean here?
We meant “similar". We corrected that sentence.
Line 300 I don’t think egregiously is the right word here.
Correct, we omitted that word.
Line 310 insert “such” before as pyridoxal
Done.
Line 339 better “the absolute concentration of Hb is an important”.
Done.
Line346 forcefully? Better necessarily.
Done.
Line 347 “addressed” better than approached, “while” better than yet.
Done.
Line 402 “elicit” better than exhibit.
Done.
Line 412 “has been” better than was.
That paragraph was rewritten.
Line 428/429 “of” better than led by.
Done.
Line 445 delete groups.
Done.
Line 462 difficult to understand the effect of encapsulation on NO but not CO. Is the NO present not as the gas but as one of nitrogen compounds such as mentioned in Fig 2?
Yes, we have clarified the concept as follows:
Encapsulation of concentrated Hb in phospholipid vesicles with diameters from 50 to 8000 nm retards the reaction of Hb with NO, but not with CO, due to a resistance attributed to the intracellular diffusion barrier [182]. Such resistance may be associated with the transformation of gaseous NO into inorganic salts as shown in Figure 2, while CO reacts with Hb only as a gas.

Reviewer 3 Report
This is an excellemt review of the role of NO. However, this is a very complicated problem and the authors have overemphasize the role of NO.
NO plays an important role but many other factors are also important. The author did mention these very briefly, but they give the impression that NO is the most important single factor. The authors could make it very clear in the title and in the abstract and introduction that this review is on the role of NO and not on other important factors. They have done an excellent review on NO but gave the wrong impression that NO is the single most important factor
Author Response
Reviewer 3- Round 1
This is an excellemt review of the role of NO. However, this is a very complicated problem and the authors have overemphasize the role of NO.
NO plays an important role but many other factors are also important. The author did mention these very briefly, but they give the impression that NO is the most important single factor. The authors could make it very clear in the title and in the abstract and introduction that this review is on the role of NO and not on other important factors. They have done an excellent review on NO but gave the wrong impression that NO is the single most important factor.
We appreciate that the Reviewer shared with us the impression that in this manuscript we have over-emphasized the role of NO in the development of HBOC. Therefore, to soften this impression, we changed the text in three parts:
The last sentence in the first paragraph of the introduction:
Among the plethora of side effects that have emerged from time to time, the fast reaction of acellular Hb with nitric oxide (NO) represents a serious and yet unsolved mechanism that prevented and is still preventing the launch of affordable HBOC.
The last sentence in the last paragraph of the introduction:
Thus, the purpose of this article in the special issue dedicated to the 25th anniversary of NO is not to add a further review of past and present HBOC candidates but rather to focus on the basic biochemical and physiological events that hindered the launch of a single HBOC candidate in the clinical arena, with special concern to one of the several interactions of Hb with other molecules: the reaction with NO. We limited our literature search to peer-reviewed articles excluding abstracts, patents, book chapters, and other non-reviewed reports. We apologize for not citing several articles for space reasons
The second sentence of the conclusion:
Many bottlenecks have occurred. While most of them have been solved, or are being solved by various protein engineering approaches, the fast binding of NO with Hb appears one of the barriers that are considerably hindered the launch of these life-saving therapeutics.
We hope that now the wrong impression we gave in the first version of the manuscript has vanished.

Round 2
Reviewer 3 Report
This is an excellent review on nitric oxide and the authors have made most of the requested revision. HOWEVER, in the present revised form,in order to be more up to date and represent more views on other aspects of the topic they should not just refer to the 2006 book by Winslow but should include the more recent 2021 OPEN ACCESS book on Blood Substitute edited by the past and present presidents of ISBS ( Chang, Bulow,Jahr, Sakai and Yang) on Nanobiotherapeutic basis for blood substitutes:
www.medicine.mcgill.ca/artcell/2021book.pdfOnce they do this, the article can be accepted for publication
Author Response
Reply to Reviewer 3
This is an excellent review on nitric oxide and the authors have made most of the requested revision. HOWEVER, in the present revised form,in order to be more up to date and represent more views on other aspects of the topic they should not just refer to the 2006 book by Winslow but should include the more recent 2021 OPEN ACCESS book on Blood Substitute edited by the past and present presidents of ISBS ( Chang, Bulow,Jahr, Sakai and Yang) on Nanobiotherapeutic basis for blood substitutes:
Once they do this, the article can be accepted for publication
For equilibrated information, we have added that reference [1] in the first paragraph of the Introduction (as reference 8 in the ms) in the correspondence of the 2006 book by Winslow, which was cited only once in the manuscript.
In addition, we have inserted in subchapter 2.4 one of the contributions as a reference [2] in a new sentence that deals with an argument that was neglected in the previous version of the manuscript, accompanied by other peer-reviewed references by the same Authors.
The critical role of NO scavenging in leading vasoconstriction was confirmed by testing the hypothesis that inhaled NO may mitigate the vasoconstrictive effects of HBOC [2]. Indeed inhaling 80 ppm one hour before HBOC infusion in mice, lambs, and sheep could prevent most of the deleterious effects of NO scavenging [2-4]. This treatment appeared to be effective also in a single compassionate use in humans [5].
- Chang, T. M. S. A. B., Leif%A Jahr, Jonathan%A Sakai, Hiromi%A Yang, Chengmin, Nanobiotherapeutic Based Blood Substitutes.
- Yu, B.; Zapol, W. M., Hemoglobin-based Oxygen Carriers and Inhaled Nitric Oxide. In Nanobiotheraputic Based Blood Substitues, Chang, T. M. S.; Bulow, L.; Jahr, J. S.; Sakai, H.; Yang, C., Eds. World Scientific: 2021; pp 267-277.
- Yu, B.; Raher, M. J.; Volpato, G. P.; Bloch, K. D.; Ichinose, F.; Zapol, W. M., Inhaled nitric oxide enables artificial blood transfusion without hypertension. Circulation 2008, 117, (15), 1982-90.
- Yu, B.; Bloch, K. D.; Zapol, W. M., Hemoglobin-based red blood cell substitutes and nitric oxide. Trends Cardiovasc Med 2009, 19, (3), 103-7.
- Marrazzo, F.; Larson, G.; Sherpa Lama, T. T.; Teggia Droghi, M.; Joyce, M.; Ichinose, F.; Watkins, M. T.; Stowell, C.; Crowley, J.; Berra, L., Inhaled nitric oxide prevents systemic and pulmonary vasoconstriction due to hemoglobin-based oxygen carrier infusion: A case report. J Crit Care 2019, 51, 213-216.